# Vascular Calcification: Key Roles of Phosphate and Pyrophosphate

**DOI:** 10.3390/ijms222413536

**Published:** 2021-12-17

**Authors:** Ricardo Villa-Bellosta

**Affiliations:** 1Center for Research in Molecular Medicine and Chronic Diseases (CiMUS), Av Barcelona. Campus Vida, Universidade de Santiago de Compostela, 15782 Santiago de Compostela, Spain; ricardo.villa@usc.es; 2Department of Biochemistry and Molecular Biology, Universidade de Santiago de Compostela, Plaza do Obradoiro s/n, 15782 Santiago de Compostela, Spain

**Keywords:** vascular calcification, pyrophosphate, phosphate, ATP, calcium

## Abstract

Cardiovascular complications due to accelerated arterial stiffening and atherosclerosis are the leading cause of morbimortality in Western society. Both pathologies are frequently associated with vascular calcification. Pathologic calcification of cardiovascular structures, or vascular calcification, is associated with several diseases (for example, genetic diseases, diabetes, and chronic kidney disease) and is a common consequence of aging. Calcium phosphate deposition, mainly in the form of hydroxyapatite, is the hallmark of vascular calcification and can occur in the medial layer of arteries (medial calcification), in the atheroma plaque (intimal calcification), and cardiac valves (heart valve calcification). Although various mechanisms have been proposed for the pathogenesis of vascular calcification, our understanding of the pathogenesis of calcification is far from complete. However, in recent years, some risk factors have been identified, including high serum phosphorus concentration (hyperphosphatemia) and defective synthesis of pyrophosphate (pyrophosphate deficiency). The balance between phosphate and pyrophosphate, strictly controlled by several genes, plays a key role in vascular calcification. This review summarizes the current knowledge concerning phosphate and pyrophosphate homeostasis, focusing on the role of extracellular pyrophosphate metabolism in aortic smooth muscle cells and macrophages.

## 1. Introduction

Pathologic calcification of cardiovascular structures, or vascular calcification, is associated with several diseases (for example, genetic diseases, diabetes, and chronic kidney disease), and is a common consequence of aging [1,2]. Vascular calcification, the deposition of phosphate-calcium crystals on the cardiovascular system, mainly in blood vessels, myocardium, and cardiac valves, is one of the most important factors determining patients’ morbidity and mortality worldwide [3].

In blood vessels, calcified deposits are found in distinct layers of the aortic wall and are associated with specific pathologies. Intimal calcification occurs in atherosclerotic lesions and is associated with vascular smooth muscle cells and macrophages [4]; whereas medial calcification (so-called “Monckeberg’s medial sclerosis”) occurs in the medial layer of the aortic wall and is associated with the collagen/elastin fibers and vascular smooth muscle cells [5,6].

Different mechanisms regarding the pathogenesis of vascular calcification have been proposed [2], including (1) loss of inhibitions, (2) calcium and phosphorus homeostasis, (3) osteochondrogenic differentiation of vascular cells, (4) apoptosis, (5) circulating nucleation complexes/paracrine factors, and (6) matrix degradation. However, despite major advances in recent years, our understanding of calcification pathogenesis is far from complete.

## 2. Role of Phosphate

Inorganic phosphate is essential for a variety of cellular processes, such as energy metabolism, bone formation, and synthesis of biomolecules, including phospholipids and nucleic acids. However, elevated serum phosphorus (in the form of inorganic phosphate) has emerged as a key risk factor for vascular calcification [7,8]. During the past decade, in vitro experiments have shown calcium-phosphate deposits in vascular smooth muscle cells incubated with high phosphate concentration [8]. Logically, this observation was first interpreted as the consequence of an increase in phosphate transport, with the consequent increase in the intracellular phosphate concentration [9]. However, several studies show that phosphate transporters are saturated with normal serum phosphate levels [10,11]. Moreover, additional studies show that the formation of calcium-phosphate crystals is a passive physicochemical process that does not require any cellular activity, suggesting an important role of phosphate homeostasis [6,11,12]. Notably, there are two major consequences regarding the fate of vascular smooth muscle cells in phosphate-induced vascular calcification. The first involves apoptosis-dependent matrix mineralization, which has been detected both in cultured human vascular smooth muscle cells and in arteries from pediatric dialysis patients [13,14,15]. The second invokes a profound transition to a bone-forming phenotype [16]. In support of this notion, in vitro studies have shown that elevated phosphate results in the expression of osteochondrogenic markers (such as BMP-2 and Runx2/Cbfa1, a transcription factor that induces the expression of major components of the bone matrix) [17,18]. However, recent studies show that calcium-phosphate deposits can induce both the transition to a bone-forming phenotype and apoptosis in vascular smooth muscle cells and the aortic wall, suggesting that the active mechanisms described could be a response to the excessive formation and deposit of calcium-phosphate crystals [5,6,19,20].

### 2.1. Biomineralization Process

The formation and deposition of inorganic minerals within or outside the cells of various organisms is known as biological mineralization or biomineralization. Biomineralization in hard tissues (such as in bone or dentine) is normally considered a physiological process; however, the accumulation of inorganic minerals in soft tissues (such as blood vessels, joints, and internal organs, including muscle, liver, or brain) is considered pathological or ectopic biomineralization. Under normal conditions, the soft tissues are not mineralized, but due to aging and other pathological conditions, soft tissues become calcified, which leads to morbidity and mortality. The main biominerals found in mineralized vertebrate connective tissue are calcium-phosphate salts. In an aqueous system of calcium and phosphate, there are several known non-ion-substituted calcium phosphates, which have also been found in calcified tissues. The phosphate ion is a central phosphorus atom surrounded by four oxygen atoms in a tetrahedral arrangement. In biological systems, it is found as a free phosphate ion in solution (inorganic phosphate) or bound with different biological molecules, including proteins, sugars, lipids, and nucleic acid. Aqueous inorganic phosphate exists in four forms according to its triprotic equilibrium: (1) trihydrogen phosphate ion (H_3_P0_4_), (2) dihydrogen phosphate ion (H_2_PO_4_^−^), (3) hydrogen phosphate ion (HPO_4_^2−^), and (4) phosphate ion (PO_4_^2−^); (see Figure 1). Inorganic phosphate is quite strong with respect to the first dissociation (pK_a1_ = 2.1), moderately weak with respect to the second (pK_a2_ = 6.9), and very weak with respect to the third (pK_a3_ = 12.4). Under strongly basic or acidic conditions, the phosphate ion or trihydrogen phosphate dominates, respectively. In extracellular fluid (pH = 7.4), only H_2_PO_4_^−^ and HPO_4_^2−^ ions are present in significant amounts in a proportion of 1:4, respectively. Whereas, in cytosol (pH = 7) and lysosome (pH = 4.8), this proportion is inverted (1.6:1 and 99:1, respectively).

Notably, various phosphates-calcium salts are obtained by charge neutralizing these different inorganic phosphate ions in the presence of a calcium ion [21], including (1) monocalcium phosphate anhydrous (Ca(H_2_PO_4_)_2_), (2) dicalcium phosphate anhydrous (CaHPO_4_), and (3) β-tricalcium phosphate (β -Ca_3_(PO_4_)_2_) [22,23]. Both Ca(H_2_PO_4_)_2_ and CaHPO_4_ are hydrated to form their hydrated forms (monocalcium phosphate monohydrate (Ca(H_2_PO_4_)_2_H_2_O) and dicalcium phosphate dihydrate (CaHPO_4_2H_2_O). CaHPO_4_2H_2_O, also called Brushite, is often found in calcified tissues, whereas Ca(H_2_PO_4_)_2_, Ca(H_2_PO_4_)_2_H_2_O, CaHPO_4_, and β -Ca_3_(PO_4_)_2_ have never been found in calcifications [23]. The Mg-substituted β-tricalcium phosphate form (whitlockite) is not formed under physiological conditions. However, whitlockite was also found in some calcified tissues, such as in the aorta in hemodialysis patients [24,25].

The final product of the calcium-phosphate salts’ reaction in neutral or basic solutions is crystalline hydroxyapatite (Ca_10_(PO_4_)_6_(OH)), the main component of bone and calcified tissues; along with two of its precursors (amorphous calcium phosphate, Ca_9_(PO_4_)_6_nH_2_O, and octocalcium phosphate, Ca_8_H_2_(PO_4_)_6_5H_2_O) [22,26]. Notably, amorphous calcium phosphate, which is also found in soft-tissue pathological calcifications, consists mainly of roughly spherical Ca_9_(PO_4_)_6_ clusters (called Posner’s clusters) that appear to be energetically favored compared to (Ca_3_(PO_4_) and Ca_6_(PO_4_)_4_ clusters [21,22]. Therefore, the structure of hydroxyapatite [26] can be interpreted as an aggregation of Posner’s clusters [27,28] (Figure 1). Notably, Mg^2+^ and ATP are critical for the stabilization of amorphous calcium phosphate [29,30].

According to the charge neutralization theory of calcification [31], the high glycine content of elastin and collagen proteins favors the formation of beta-turns that are known to interact with calcium ions. Therefore, the deposition of these calcium-phosphate salts, both in vitro and in vivo, takes place on these extracellular matrix proteins [5,6]. For example, in bone and connective tissues, these salts are predominantly deposited on the elastic and type I collagen fibers. Moreover, in the aorta wall, calcium-phosphate crystals are deposited on elastin, the main component of the elastic fibers in the medial layer [6]. Notably, a study showed that a mouse model of elastin haploinsufficiency exhibited a significant reduction in arterial calcification [32]. In contrast, phosphate-induced mineralization, both in vitro and in vivo, is accelerated by the products derived from elastin degradation [33].

Finally, the depositions of calcium-phosphate crystals in soft tissues can be classified into three main categories: (1) calcinosis, (2) dystrophic calcification, and (3) metastatic calcification. In the presence of normal homeostasis of phosphate, calcinosis and/or dystrophic calcification occur, most often in subcutaneous tissues, skin, and related connective tissues, whereas, in the second case, calcification occurs in degenerated or necrotic tissue. Metastatic calcification occurs in normal tissues when the calcium levels are elevated in serum.

### 2.2. Phosphate Homeostasis

In adults, normal phosphate concentration in serum or plasma is mainly 2.5 to 4.5 mg/dL (0.81 to 1.45 mmol/L). Elevated serum phosphate (hyperphosphatemia) is a key risk factor for pathologic calcification in cardiovascular structures [1]. Treatment of hyperphosphatemia with phosphate binders is associated with the slow progression of cardiovascular calcification in hemodialysis patients [34]. Therefore, the homeostasis of phosphate plays a critical role in the initiation and progression of calcification [35] (see Figure 2).

Many different foods contain phosphorus, including vegetables, grains, legumes, eggs, fish, and meats. In addition, phosphate additives such as phosphoric acid, sodium phosphate, and sodium polyphosphate are present in many processed food products. Phosphate deficiency is rare in Europe, and is rarely the result of low dietary intakes. However, phosphate is also available in dietary supplements containing only phosphate o supplements containing phosphate in combination with other ingredients, including vitamins and minerals.

Recommended dietary daily phosphorus intake in healthy adults (>18 years old) is 700 mg. However, daily phosphorus intake varies between 700 and 2000 mg. After absorption, phosphorus is transported across cell membranes as phosphate. And in extracellular fluid (including plasma), phosphate undergoes one of three fates: (1) elimination, mainly by the kidneys, (2) transport into cells, or (3) deposition in bone or soft tissue (see Figure 2).

In healthy adults, oral phosphate intake is balanced mainly by phosphate excretion in the urine and feces. In this case, different factors play an important role in the control of phosphate homeostasis, including phosphate excretion and absorption by the kidneys, intestines, and bone. Although the kidney is the major regulator of phosphate homeostasis, plasma phosphate levels are altered by intestinal phosphate absorption. Notably, in normal adults, between 75% and 85% of the daily phosphate filtered by the glomerulus is reabsorbed by the renal tubules (mainly the proximal tubule) [36,37].

An increased absorption or decreased phosphate excretion can induce a relatively small elevation in serum phosphate, which has been correlated with the presence of calcified vessels due to an increase in calcium-phosphate crystal formation and saturation in the inhibition. Several diseases have been correlated with the dysregulation of phosphate homeostasis, including osteoporosis, diabetes mellitus, hyperparathyroidism, vitamin D (hyper-and hypovitaminosis), and chronic renal disease [38].

### 2.3. Phosphate Transporters

Cellular phosphate levels are controlled by sodium-phosphate co-transporters (NaPi) [21,39]. The roles of sodium-phosphate cotransporters in human clinical disease and physiology processes have not been yet well defined. Two families of sodium-phosphate cotransporters have been principally identified, each with multiple members: Type II (also called SLC34 or NaPi-II) and type III (SLC20 or NaPi-III), which transport phosphate with high affinity (K_m_ ≈ 0, 1 or less) but show differences in their affinities for H_2_PO_4_^−^ and HPO_4_^2−^ ions [40,41]. Originally identified as a phosphate transporter, Type I (SLC17 or NaPi-I) phosphate transporters mediate the transmembrane transport of organic anions, with relativity low affinity for phosphate suggesting that they transported organic and inorganic anions more readily than phosphate [42].

The SCL34 family comprises three members (also called NaPi-II), which are expressed in the small intestine (NaPi-IIb) and the kidney (NaPi-IIa and NaPi-IIc), two important sites for the control of phosphate homeostasis [43]. NaPi-IIa is expressed predominantly in renal proximal tubules, and under normal conditions, NaPiIIa is the transporter responsible for 95% of phosphate reabsorption in the proximal tubule. An expression of NaPi-IIc was found exclusively in the kidney and described as being growth-related [44,45]. Moreover, the SLC20 family of solute carriers are represented by Pit-1 and Pit-2 (Type III sodium-phosphate cotransporters) [46]. Both cotransporters mediate the movement of phosphate ions across the cell membrane and are ubiquitously expressed, suggesting a “housekeeping” function. More precise localization studies revealed different levels of Pit-1 and Pit-2 expression in each cell type [10].

The roles of sodium phosphate cotransporters in pathophysiology have not been well defined, but their important role in controlling phosphate homeostasis and intracellular phosphate levels for the synthesis of macromolecules and energy metabolism make them an important target to study.

## 3. Role of Pyrophosphate

In vertebrates, plasma and other extracellular fluids are supersaturated with phosphate and calcium, causing a tendency for spontaneous calcium-phosphate precipitation [5,6,12] (see Figure 1). Therefore, the synthesis of calcium phosphate deposition inhibitors is essential for survival, including pyrophosphate and several proteins (such as Matrix Gla Protein, Fetuin-A and osteopontin) [4].

Matrix Gla protein is a mineral-binding extracellular matrix protein synthesized mainly by vascular smooth muscle cells and chondrocytes, the first protein recognized as an inhibitor of vascular calcification in vitro and in vivo [47]. Matrix Gla protein contains several Vitamin K-dependent carboxylation/gamma-carboxyglutamic (Gla) amino acid residues, which are responsible for the high-affinity binding of calcium ions. Notably, matrix Gla protein-deficient mice exhibit spontaneous calcification of the arteries and cartilage, and several studies reported possible associations between plasma Matrix Gla protein and vascular calcification in uremic, diabetic, atherosclerotic, and hypertensive patients. Matrix Gla protein is present in atherosclerotic lesions.

Fetuin-A is a circulating plasma glycoprotein that also has the capacity to bind calcium and has anti-inflammatory properties [48]. Notably, Fetuin-A knockout-mice spontaneously develop soft tissue calcification of the heart, vessels, kidney, testis, and skin [49,50]. However, the relationship between serum fetuin-A levels and vascular calcification remains unclear [51].

Osteopontin, a sialic acid-rich glycoprotein first purified from the bone, is a known noncollagenous bone matrix protein that regulates calcification [7,52]. Like Matrix Gla Protein, osteopontin also regulates calcification during bone development and remodeling. Moreover, osteopontin is also expressed by macrophages, smooth muscle, and endothelial cells in human aortic and coronary atherosclerotic plaques [53,54]. However, upregulation of osteopontin mRNA levels is not correlated with vascular calcification, suggesting that osteopontin might not be necessary for calcification [51]. In addition, several studies suggest that osteopontin is not an endogenous inhibitor of calcification in the aortic wall [55,56].

On the other hand, extracellular pyrophosphate is the major endogenous physicochemical inhibitor of calcium-phosphate crystal formation and growth, both in vitro and in vivo [57]. Extracellular pyrophosphate acts by avidly binding to nascent hydroxyapatite crystals with complete inhibition at micromolar concentration [5,6,11], which is more than 1000-fold less than physiologic calcium or phosphate concentrations. Extracellular pyrophosphate is present at levels sufficient to completely prevent hydroxyapatite formation of physiologic calcium or phosphate concentrations [11]. However, loss of extracellular pyrophosphate synthesis or increments of plasmatic phosphate concentration (hyperphosphatemia) lead to vascular calcification due to a lack of inhibitory capacity [35]. For example, studies have shown that plasma pyrophosphate is reduced after standard hemodialysis in a mouse model of progeria [34,58]. Consequently, several studies show that daily injections of exogenous pyrophosphate prevent medial vascular calcification in experimental rat and mice models, including progeria and renal failure [59,60,61]. Notably, several therapeutic strategies that increase endogenous extracellular pyrophosphate synthesis prevent the excessive vascular calcification found in the medial layer of the aortic wall in progeria mice [62,63]. Therefore, pyrophosphate deficiency [64] is a critical risk factor for vascular calcification, suggesting an important role of pyrophosphate homeostasis and extracellular pyrophosphate metabolism in vascular calcification [35].

### 3.1. Extracelular Pyrophosphate Metabolism

The currently known enzymes and transporters involved in extracellular pyrophosphate metabolism include members of the ecto-nucleotide pyrophosphatase/phosphodiesterase, tissue-nonspecific alkaline phosphatase, ecto-5′-nucleotidase, equilibrative nucleoside transporters, phosphate transporters (NaPi), progressive ankylosis proteins, and pump/channels that release ATP extracellularly, including the multi-drug resistance-associated protein 6 [21,64,65,66]. Therefore, understanding the role of enzymes and transporters involved in the extracellular pyrophosphate metabolism could provide potential future therapeutic targets to prevent vascular calcification (see Figure 3).

Pyrophosphate is mainly produced during the extracellular hydrolysis of ATP [6,67,68]. The major generator of endogenous extracellular pyrophosphate in several tissues, including the aorta, is the enzyme ecto-nucleotide pyrophosphatase/phosphodiesterase (eNPP), which hydrolyzes extracellular ATP to generate pyrophosphate and AMP [67] (see Figure 3). Three members of the eNPP activity have been found (eNPP1-3); they exist both as membrane proteins, with an extracellular active site, and as soluble proteins in body fluids (also known as PC-1, autotaxin, and B10, respectively) [65]. In aorta and vascular smooth muscle cells, eNPP1 is the main source of extracellular pyrophosphate [6,60,68]. Mutations in eNPP1 result in generalized arterial calcification of infancy (see Table 1), characterized by an excessive calcification of the internal elastic lamina of large and medium-sized arteries [67]. Moreover, eNPP1-null mice develop ectopic artery calcification [69]. Notably, ATP is also a direct inhibitor of calcification [70], with a physicochemical mechanism similar to pyrophosphate, bisphosphonates (non-hydrolyzable analogous of pyrophosphate), and polyphosphates [57,71].

Moreover, pyrophosphate is degraded to phosphate mainly by tissue non-specific alkaline phosphatase (TNAP) in tissues and extracellular fluids (see Figure 3). Cells over-expressing TNAP, or the addition of alkaline phosphatase in culture media, is sufficient to cause medial vascular calcification in the aortic ring ex vivo [68,72]. Notably, TNAP activity is increased in models of medial vascular calcification, such as in uremic rats or in a mouse model of progeria [34,60,73]. Additionally, several studies have shown that phosphatase inhibitors can prevent vascular smooth muscle calcification in vitro and in vivo and that the ablation of phosphatase function produces a loss of skeletal mineralization [74,75]. TNAP is a non-specific ecto-phosphomonoesterase and a GPI-anchored membrane enzyme, with an extracellular active site and a soluble protein in body fluids. It releases phosphate from various organic compounds, including pyrophosphate [65,75].

Another enzyme involved in vascular calcification is the membrane-bound ecto-5′nucletotidase (5NT, NT5E, or CD73), which preferentially binds AMP and converts it to adenosine and phosphate (see Figure 3). Mutations in ecto-5′nucletotidase induce medial arterial calcification of the lower extremity arteries with peri-articular calcification [76]. Like TNAP, ecto-5′nucletotidase is a GPI-anchored enzyme with an extracellular active site and a soluble form cleaved from GPI-anchor. Moreover, like phosphate, adenosine should be recovered from the extracellular space to generate ATP by mitochondria or another metabolic pathway [21]. Notably, the first report of a role for adenosine transport in regulating the calcification of soft tissues shows that the loss of the equilibrative nucleoside transporter 1 (ENT1, Slc29a1) in mice could explain the diffuse idiopathic skeletal hyperostosis in humans, characterized by the ectopic calcification of spinal tissues [77] (see Figure 3). In addition, impaired synthesis of intracellular ATP due to mitochondrial dysfunction has been associated with a reduction in extracellular pyrophosphate concentration, as well as vascular calcification, in a mouse model of premature aging [60]. A recent study showed that magnesium treatment improved mitochondrial ATP synthesis and reduced vascular calcification in this mouse model [63].

Finally, in 2000 two additional new genes were identified that could play an important role in controlling tissue calcification and arthritis: progressive ankylosis protein and multidrug resistance-associated protein 6. However, the molecular mechanisms remain in part unknown. First, it was reported that mutations in the progressive ankylosis gene cause a severe form of generalized joint calcification and arthritis [78]. Loss of progressive ankylosis function causes excessive hydroxyapatite formation in progressive ankylosis gene null mice [78]. Overexpression of progressive ankylosis protein in cultured tissue cells increases extracellular pyrophosphate, and cells from the progressive ankylosis protein mutant have a reduction in extracellular pyrophosphate levels [78]. In a first study, 7–12 membrane-spanning helices and a central channel for the progressive ankylosis protein [78] was proposed. Consequently, it seems as though the progressive ankylosis channel regulates pyrophosphate transport from the cytoplasm to the extracellular milieu; however, additional studies showed that progressive ankylosis protein could be a channel or regulator of adjacent channels which release ATP outside the cells. Notably, in humans, mutations in the channel core of progressive ankylosis protein cause craniometaphyseal dysplasia, a rare skeletal condition of abnormal bone formation characterized by an increased density of craniofacial bones and abnormal modeling of the metaphysis of the tubular bones [79,80]. Moreover, mutations in the N- and C-terminus of the progressive ankylosis protein cause chondrocalcinosis, a disease of articular cartilage that is radiographically characterized by the deposition of calcium pyrophosphate dihydrate crystals in the joints [81,82,83]. Craniometaphyseal dysplasia is associated with decreased extracellular pyrophosphate levels, whereas chondrocalcinosis is associated with an increase in the amount of pyrophosphate in the extracellular space, which induces the spontaneous formation of calcium pyrophosphate crystals.

Phosphate and pyrophosphate concentration (and, therefore, the phosphate/pyrophosphate ratio), known risk factors for vascular calcification, is strictly controlled by a complex interplay of genes [35]. Progressive ankylosis gene and protein could play a key role in this complex process by regulating both eNPP1 and TNAP activities and ATP excretion by different channels. In support of this suggestion, the over-expression of wild-type progressive ankylosis protein results in down-regulation of TNAP activity in chondrogenic cells, and transfection of eNPP1 in osteoblasts enhance extracellular pyrophosphate levels only when wild-type progressive ankylosis protein is present [64].

The second-gen reported is multidrug resistance-associated protein 6 (MRP6), also known as ATP-binding cassette sub-family C member 6 (ABCC6) [84,85]. It was reported that mutations in this gene cause *Pseudoxanthoma elasticum*, a heritable disorder of connective tissue characterized by calcification of the elastic fibers in skin, arteries, and retina. MRP6/ABCC6 is a member of the superfamily of ABC transporters, composed of several related pumps that can transport various molecules across extra- and intra-cellular membranes, including glutathione-S-conjugates and cyclic nucleotides. This suggests that MPR6/ABCC6 may act as a pump that releases endogenous, low molecular weight inhibitors of calcium phosphate deposits in fluids outside cells, such as ATP or citrate; however, this has not been thoroughly demonstrated.

### 3.2. Extracelular Pyrophosphate Metabolism in the Aortic Wall

In the healthy aortic wall, the pyrophosphate synthesis from ATP hydrolysis is several times faster than pyrophosphate hydrolysis [6,68]. Since vascular smooth muscle cells are the main cell type involved in preventing medial calcification of the aortic wall, the expression and activity of TNAP and eNPP enzymes play a critical role in the prevention of medial vascular calcification [60,62,73]. Notably, both the phosphate-induced aortic and the vascular smooth muscle cells calcification processes, in vitro and ex vivo, respectively, vary depending on the stage of the calcification [6]. In the early phase, when calcification is not yet present, eNPP and TNAP activities are increased or decreased, respectively, in vascular smooth muscle cells both in vitro, ex vivo, and in vivo [6]. By contrast, in the late phase, when calcification is present and Runx2/Cbfa1 is expressed, hydroxyapatite increases both eNPP1 and TNAP activity, suggesting a compensatory increment in pyrophosphate synthesis in the early phase of phosphate-induced calcification [6].

On the other hand, calcification is a very common complication of atherosclerosis that involves aortic smooth muscle cells, monocyte infiltration, and macrophage accumulation within the artery wall [4]. In response to a variety of microenvironmental signals, including those found in different regions of atherosclerotic plaques and at distinct stages of atherosclerosis, macrophages polarize, giving rise to a phenotypically heterogeneous cell population with distinct functions [86]. Although macrophages display remarkable plasticity and can change their physiology and function in response to environmental cues, atherosclerotic lesion contains cells expressing markers of classical macrophages (M1 macrophages) and alternative macrophages (M2 macrophages), which represent the two major and opposing activities of a wide range of macrophage phenotypes. For example, M1 macrophages promote inflammation, inhibit cell proliferation, and cause tissue damage, whereas M2 macrophages promote cell proliferation and tissue repair [87].

Notably, in a macrophage/vascular smooth muscle cell in vitro co-culture system, macrophages enhance the calcifying capacity of vascular smooth muscle cells by inducing phenotypic changes, including matrix mineralization and increment in TNAP activity [88,89]. Moreover, activators of the vitamin D receptor (calcitriol and paricalcitol) promote calcification in a macrophage/vascular smooth muscle cell co-culture [90]. These findings suggest that macrophages could contribute to the calcification of the atherosclerotic plaque in vivo (see Figure 4).

In a recent study [89], the authors show that M2 macrophages also have anti-calcifying properties due predominately to their increased capacity to synthesize extracellular pyrophosphate. M2 macrophages release more ATP and increase pyrophosphate synthesis via increased eNPP1 expression and activity, compared with M1 macrophages. Moreover, a co-culture of vascular smooth muscle cells with M2 macrophages increases eNPP1 expression and activity in vascular smooth muscle cells. In contrast, a co-culture of vascular smooth muscle cells with M1 macrophages increases TNAP expression and activity in vascular smooth muscle cells [89].

Finally, a study also shows that hyperphosphatemia can activate macrophages, forming a different and new macrophage type [91]. Phosphate-induced macrophages (MPi) express M2 markers and have similar activities to M2 macrophages, including arginine degradation via arginase 1, higher metabolic activity, and increased antioxidant production (see Figure 4). Consequently, as with M2 macrophages, MPi macrophages also possess anti-calcifying properties via increased extracellular pyrophosphate availability. In contrast, calcium-phosphate crystals present in atherosclerotic lesions, including hydroxyapatite, can induce macrophage polarization into M1 macrophages [91]. These findings suggest two separate environments and steps are involved during the process of calcification in the atheroma plaque, similar to the two steps also found during medial calcification [6]. However, additional deeply studies are necessary [92].

## 4. Summary

Extracellular pyrophosphate is a major endogenous calcification inhibitor, and its metabolism plays a key role in vascular calcification. The use of pyrophosphate as a therapeutic is limited by its pharmacokinetics and the fact that stable analogs inhibit bone formation. However, strategies to raise pyrophosphate levels by targeting its metabolism may have promise as therapies in the future.

## Figures and Tables

**Figure 1 ijms-22-13536-f001:**
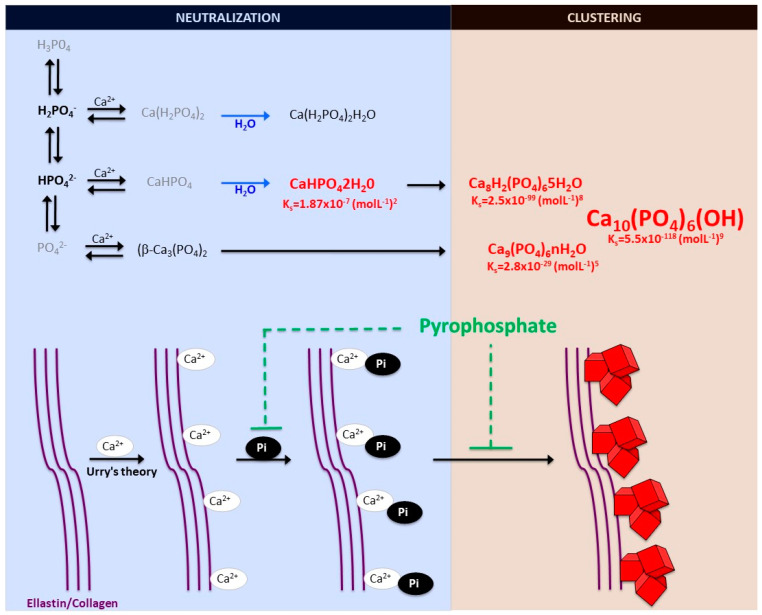
Schematic representation of calcium-phosphate crystal formation. Phosphate exists in four forms in biological systems: trihydrogen phosphate (H_3_P0_4_), dihydrogen phosphate ion (H_2_PO_4_^−^), hydrogen phosphate ion (HPO_4_^2−^), and phosphate ion (PO_4_^2−^). Various phosphate-calcium salts are produced in the presence of calcium, including anhydrous monocalcium phosphate (Ca(H_2_PO_4_)_2_), anhydrous dicalcium phosphate (CaHPO_4_), β-tricalcium phosphate (β-Ca_3_(PO_4_)_2_), monocalcium phosphate monohydrate (Ca(H_2_PO_4_)_2_H_2_O), and dicalcium phosphate dihydrate (CaHPO_4_2H_2_0, also called Brushite). The final product of the calcium and phosphate reaction is crystalline hydroxyapatite (Ca_10_(PO_4_)_6_(OH)), the main component of bone and calcified tissues and two of its precursors, amorphous calcium phosphate (Ca_9_(PO_4_)_6_nH_2_O) and octocalcium phosphate (Ca_8_H_2_(PO_4_)_6_5H_2_O). Pyrophosphate directly inhibits the formation and growth of phosphate-calcium crystals, mainly hydroxyapatite.

**Figure 2 ijms-22-13536-f002:**
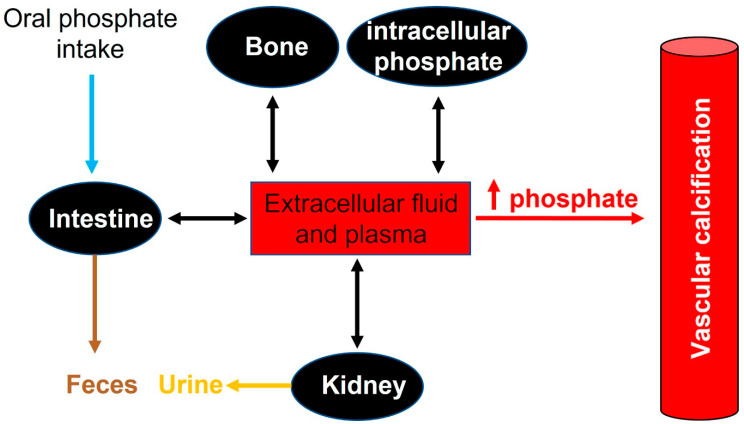
Phosphate flux between body compartments. Phosphate balance is a complex process involving bone intestinal absorption and dietary phosphate and renal excretion of phosphate.

**Figure 3 ijms-22-13536-f003:**
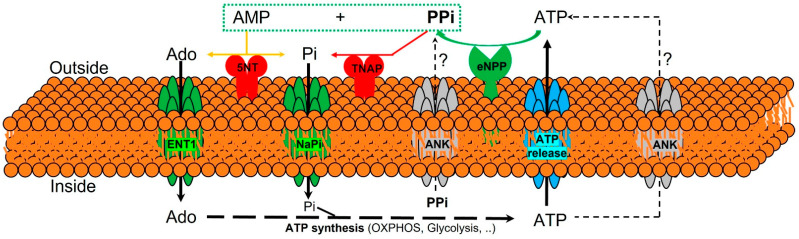
Schematic representation of the ectoenzymes and transporters involved in the extracellular pyrophosphate metabolism. Ectonucleotide pyrophosphatase phosphodiesterase (eNPP) hydrolyze ATP releasing pyrophosphate (PPi) and adenosine-5′-monophosphate (AMP). Pyrophosphate is degraded to phosphate (Pi) by tissue non-specific alkaline phosphatase (TNAP). ATP is released by cells via exocytotic mechanisms and via multiple types of membrane channels, including ABCC6. The progressive ankylosis (ANK) protein can contribute to extracellular pyrophosphate by transporting either ATP or pyrophosphate. Equilibrative nucleoside transporter 1 (ENT1). Sodium-phosphate co-transporter (NaPi). Ecto-5′nucletotidase (5NT). Oxidative phosphorylation pathway (OXPHOS).

**Figure 4 ijms-22-13536-f004:**
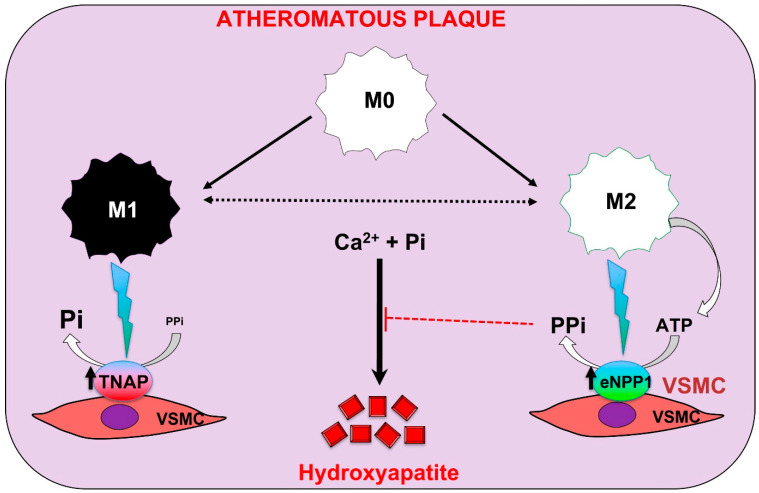
Proposed roles of different macrophage subtypes in calcification of the atheromatous plaque. Classical macrophages (M1 macrophage) induce tissue-nonspecific alkaline phosphatase (TNAP) expression in vascular smooth muscle cells (VSMCs). Moreover, the presence of alternatively macrophages (M2 macrophage) induces ectonucleoside triphosphate diphosphohydrolase 1 (eNPP1) expression in VSMCs. Pi: phosphate; PPi: pyrophosphate.

**Table 1 ijms-22-13536-t001:** Genetic Diseases involved in extracellular pyrophosphate metabolism that produces ectopic Calcification.

Genetic Disease	Ectopic Calcification	Protein Affected	Main Reference	Role
Generalized Arterial Calcification of Infancy	Medial Arterial	eNPP1	Rutsch et al., 2003	Synthesis of pyrophosphate
	Medial Arterial and Periarticular	5NT	St Hilaire et al., 2011	Hydrolysis of AMP
Idiopatic Skeletal Hypertosis	Spinal Tissues	ENT1	Warraich et al., 2013	Ado Transporter
Familial Idiopathic basal Ganglia Calcification	Basal Ganglia and cortex	Pit-2	Wang et al., 2012	Phosphate Transporter
Pseudoxanthoma ellasticum	Elastic fibers in skin, arteries and retine.	ABCC6	La Seux et al., 2000 Bergen et al., 2000	ATP transporter
Craniometaphyseal dysplasia	Craniofacial Bones	ANK	Nürnberg et al., 2001	?
Condrocalcinosis	Articular cartilage	ANK	Pendleton et al., 2002	?

## Data Availability

Data not available to be shared.

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
