# Peer review of "Vascular Calcification: Key Roles of Phosphate and Pyrophosphate"

_ijms, 2021, doi:10.3390/ijms222413536_

Round 1

Reviewer 1 Report

The author summarized the current knowledge about phosphate and pyrophosphate homeostasis and their roles for vascular calcification.  This reviewer only has minor comments. 

  1. The author described “See Figure 3” in line 291 and line 300. However, no information about NT5E and ENT1 has been shown in Figure 3.
  2. There are a lot of typos and grammatical errors throughout the manuscript. For example, line 72: mineralization o biomineralization; line 180: tansporters; line 375: the authors shown that; line 383: a study has been also show that; line 383: can activates…and more.
  3. Please correct the style of references.

Author Response

The author summarized the current knowledge about phosphate and pyrophosphate homeostasis and their roles for vascular calcification.  This reviewer only has minor comments. 

  1. The author described “See Figure 3” in line 291 and line 300. However, no information about NT5E and ENT1 has been shown in Figure 3.

Response: Figure 3 has been corrected

  1. There are a lot of typos and grammatical errors throughout the manuscript. For example, line 72: mineralization o biomineralization; line 180: tansporters; line 375: the authors shown that; line 383: a study has been also show that; line 383: can activates…and more.

Response: several typos and grammatical errors have been corrected

  1. Please correct the style of references.

Response: the reference style has been corrected

Reviewer 2 Report

This manuscript is well-written short review on the key role of phosphate and pyrophosphate metabolism in the pathophysiology of vascular calcification.

I have only some suggestions that could further enhance the quality of the paper:

1. Figure 3 may also include the key players in cyclic nucleotide metabolism and transport that contribute to vascular calcification

2. Can the author add a short paragraph on the demonstrated/putative mechanisms that contribute to the direct inhibitory effet of PPi on vascular calcifciation

3. Is there any intracellular mechanisms contributing to the production and degradation of PPi (endothelial or SMC) ? Can this intracellular metabolism contribute to vascular calcification?

Author Response

This manuscript is well-written short review on the key role of phosphate and pyrophosphate metabolism in the pathophysiology of vascular calcification.

I have only some suggestions that could further enhance the quality of the paper:

  1. Figure 3 may also include the key players in cyclic nucleotide metabolism and transport that contribute to vascular calcification

Response: Additional information has been included in the Figure 3.

  1. Can the author add a short paragraph on the demonstrated/putative mechanisms that contribute to the direct inhibitory effet of PPi on vascular calcification

Response: The following sentences have been included in the manuscript: “Extracellular pyrophosphate acts by avidly binding to nascent hydroxyapatite crystals with complete inhibition at micromolar concentration, which is more than 1000-fold less than physiologic calcium or phosphate concentrations.  Extracellular pyrophosphate is present at levels sufficient to completely prevent hydroxyapatite formation at physiologic calcium or phosphate concentrations. However, loss of extracellular pyrophosphate synthesis or increment in plasmatic phosphate concentration (hyperphosphatemia) lead to vascular calcification due to lack of inhibitory capacity”.

  1. Is there any intracellular mechanisms contributing to the production and degradation of PPi (endothelial or SMC) ? Can this intracellular metabolism contribute to vascular calcification?

Response: the author is not aware of any studies linking the loss of intracellular pyrophosphate to vascular calcification. Moreover, impaired synthesis of intracellular ATP it is the intracellular mechanism currently know which can indirectly contribute with vascular calcification. The following sentences have been also included in the manuscript: “In addition, impaired synthesis of intracellular ATP due to mitochondrial dysfunction has been associated with reduction in extracellular pyrophosphate concentration, and vascular calcification, in a mouse model of premature aging. Notably, a recent study showed that magnesium treatment improved mitochondrial ATP synthesis and reduced vascular calcification in this mouse model.”